# Considerations of COVID-19 in Ophthalmology

**DOI:** 10.3390/microorganisms11092220

**Published:** 2023-08-31

**Authors:** Maria Letizia Salvetat, Mutali Musa, Francesco Pellegrini, Carlo Salati, Leopoldo Spadea, Marco Zeppieri

**Affiliations:** 1Department of Ophthalmology, Azienda Sanitaria Friuli Occidentale, 33170 Pordenone, Italy; 2Department of Optometry, University of Benin, Benin City 300238, Edo State, Nigeria; 3Department of Ophthalmology, University Hospital of Udine, 33100 Udine, Italy; 4Eye Clinic, Policlinico Umberto I, “Sapienza” University of Rome, 00142 Rome, Italy

**Keywords:** pandemic, COVID-19, SARS-CoV-2, lockdown, ophthalmology, viral infection, vaccination, viral transmission

## Abstract

Since its emergence in early 2020, the SARS-CoV-2 infection has had a significant impact on the entire eye care system. Ophthalmologists have been categorized as a high-risk group for contracting the virus due to the belief that the eye may be a site of inoculation and transmission of the SARS-CoV-2 infection. As a result, clinical ophthalmologists, optometrists, and eyecare professionals have had to familiarize themselves with the ocular manifestations of COVID-19, as well as its treatments and vaccines. The implementation of measures to prevent the transmission of the virus, such as restrictions, lockdowns, telemedicine, and artificial intelligence (AI), have led to substantial and potentially irreversible changes in routine clinical practice, education, and research. This has resulted in the emergence of a new mode of managing patients in a routine clinical setting. This brief review aims to provide an overview of various aspects of COVID-19 in ophthalmology, including the ocular manifestations related to the disease, the modes of transmission of SARS-CoV-2 infection, precautions taken in ophthalmic practice to prevent the spread of the virus, drugs, and vaccines used in the treatment of COVID-19, the impact of the pandemic on patients, clinicians, and the eye care system as a whole, and the future of ophthalmology conditioned by this global pandemic experience.

## 1. Introduction

The COVID-19 pandemic caused by the SARS-CoV-2 virus has had a profound impact on all aspects of healthcare, including ophthalmology. As a highly infectious disease, COVID-19 has led to significant changes in the way ophthalmic care is provided, with an increased focus on infection control measures and minimizing patient contact. In addition, COVID-19 has also been found to have ophthalmic manifestations [1]. These include conjunctivitis, uveitis, retinal abnormalities, and other ocular manifestations [2,3]. Understanding the impact of COVID-19 on ophthalmology is crucial in providing safe and effective care to patients during this challenging time, which is the aim of this brief review.

Coronaviruses are a large family of viruses that can cause illness in animals and humans [4]. They are named for their crown-like spikes on their surface. Coronaviruses can cause a range of illnesses, from the common cold to more severe diseases such as severe acute respiratory syndrome (SARS), Middle East respiratory syndrome (MERS), and COVID-19 [5,6]. Coronaviruses are spread mainly through respiratory droplets when an infected person coughs, sneezes, talks, or breathes [7]. They can also be spread by touching a surface contaminated with the virus and then touching your mouth, nose, or eyes [8].

The symptoms of a coronavirus infection can range from mild to severe, including fever, cough, shortness of breath, fatigue, body aches, loss of taste or smell, and sore throat [9,10,11]. Some people may not have any symptoms at all. 

The SARS-CoV-2 is a new type of Coronavirus, first detected at the end of 2019 in China, which was responsible for the COVID-19 pandemic. There are many variants of the COVID-19 virus, which have emerged as the virus spreads and replicates in different parts of the world. Some of the most notable variants are listed in Table 1.

There are also several other variants. It is important to note that viruses mutate over time and new variants are expected to emerge. These variants are multifaceted and are usually a mix of multiple existing mutated viruses with an ability to be more, or less virulent [18]. This highlights the importance of continued efforts to prevent the spread of the virus through measures such as vaccination, wearing masks, and practicing physical distancing.

While the previous outbreaks of Coronaviruses that occurred before the COVID-19 pandemic were largely isolated in the gastric and respiratory systems [19], COVID-19 has been successfully isolated from human tears [20], with great implications for ophthalmology.

## 2. Isolation of COVID-19 Virus from Ocular Tissues

Clinical detection of COVID-19 requires isolation of the virus RNA using polymerase chain reaction (PCR) tests [21,22]. Initial body specimens in which the virus was detected include nasal and throat-sourced fluids, saliva, and blood [23]. These are also the tissues with the highest concentrations of the COVID-19 virus [24]. The presence of IgM and IgG antibodies in the blood later became an efficient widespread alternative to the PCR test [25].

For SARS-CoV-2 virus to infect a cell, the ACE-2 and CD147 receptors must be available in addition to an entry system, which is usually the TPS2 and/or cathepsin L [26,27].

The outermost layers of the ocular system, the conjunctiva, and the tear film, have been reported to carry the COVID-19 virus [28]. Some authors have however pointed to the extremely low prevalence levels of detection in tears [29] and debate on whether such levels are virulent [30]. There is however a clear knowledge that the COVID-19 virus can replicate in anterior surface tissue culture [31].

Koo et al. were able to isolate the virus in post-mortem samples of the aqueous humor [32]. Also, similar studies confirmed the isolation of the COVID-19 virus RNA in the tear film as collected using Schirmer strips [33,34] and the aqueous humor of asymptomatic patients [32]. Similar results have now been seen in the vitreous humor [35]. The reported probability of obtaining a positive result from tears and conjunctiva-related specimens among COVID-19 patients using the RT-PCR test kits ranged between 0–57%. The presence of inflamed conjunctiva and increased severity of COVID-19 symptoms were linked to a higher likelihood of isolating the virus from the tears [32,36,37]. Possible reasons for these variations in positive test rates include sampling methodology and time of the day, equipment utilized and strain of the virus isolated. Sopp and Sharda carried out a meta-analysis and calculated the pooled detection rate of COVID-19 in human conjunctiva samples and tears to be 2.7% (CI 1.4–4.8%) [34]. The authors could not find any literature on the isolation of COVID-19 in the crystalline lens or retina. 

## 3. Ocular Manifestations of COVID-19

There are multiple presentations of ocular complications resulting from and occurring alongside COVID-19. These can further be classified as ocular complications and confirmed ocular diseases.

### 3.1. Ocular Complications of COVID-19

There have been a limited number of reports on the ocular manifestations of COVID-19, and more research is needed to fully understand the relationship between the virus and the eyes. However, some examples of the reported ocular manifestations include the following:

Conjunctivitis and other inflammations of eyes’ outer layers: Considering all cases of COVID-19 disease, conjunctivitis was reported in 5-9% of cases, episcleritis in 2.2% of cases, keratitis in 0.5% of cases, and blepharitis or lid margin hyperemia in 5% of cases. Concerning the physiological mechanisms that occur during infection, these COVID-19-related ocular manifestations have been suggested to result from a direct impact of the virus on ocular tissues [38,39]. One study found that conjunctivitis was present in 1.1% of hospitalized COVID-19 patients [40]. Another study found that conjunctivitis was present in 3.6% of COVID-19 patients in China [41].

Epiphora (excessive tearing): A case report described a patient with COVID-19 who experienced epiphora in one eye, which resolved after treatment with antiviral medication. Another study by Scalinci and Trovato described epiphora as a manifestation of conjunctivitis in COVID-19 patients [42].

Dry Eye: Dry eye disease was reported in high rates among subjects affected by COVID-19 disease. School-based research revealed a preponderance of dry eye as measured with a validated paper-based instrument [43]. Female gender and being of an older age were indicated as exacerbating risk factors [44]. A theoretical explanation was proffered to be the increased screen time during the pandemic period. Diskmetas et al. also showed a correlation between mask use and reduced tear break-up time during the heat of the pandemic [45]. Researchers suggested that dry eye in itself was an epidemic during the COVID-19 pandemic [46].

Eye pain: COVID-19 sufferers may experience eye discomfort as a dull aching, soreness, or a sense of pressure. It is thought to be caused by viral infection-induced inflammation of the conjunctiva (conjunctivitis) or other ocular tissues. Several research and case reports have established the existence of ocular discomfort in COVID-19 instances. For example, Wu et al. discovered that roughly 12% of COVID-19 patients exhibited ocular symptoms such as eye discomfort [40]. Also, a case report described a patient with COVID-19 who experienced eye pain, which was relieved with topical steroids [47]. These findings imply that eye discomfort may be an additional manifestation of COVID-19, emphasizing the necessity of evaluating ocular symptoms in the disease’s diagnosis and therapy. More study is needed to thoroughly understand the underlying causes and incidence of ocular discomfort in COVID-19 individuals.

Photophobia: A few case reports have described COVID-19 patients who experienced photophobia [40,42].

Retinal changes: Retinal ganglion cells and plexiform layers were shown to have characteristic lesions when analyzed by ocular coherence tomography in a Brazilian study of 11 COVID-19 patients [48]. These findings mirrored animal model studies that reported similar changes [49].

It is worth noting that these ocular manifestations are relatively rare, and COVID-19 primarily affects the respiratory system. In most cases, the ocular symptoms linked to COVID-19 in children tend to have a mild course, and complete visual recovery is typical [50]. However, healthcare professionals should be aware of the potential ocular manifestations of the virus, particularly in patients who present with eye symptoms and have a history of exposure to COVID-19.

### 3.2. Confirmed Ocular Diseases in COVID-19

Dry eye diseases are the most common reported ocular complications of COVID-19 [51]. Krolo et al. reported that face mask use was related to higher ocular surface disease index (OSDI) scores [44]. Other possible reasons may include increased screen time occasioned by working from home and inability to obtained needed dry eye medications due to the lockdown [52].

Children with COVID-19 have been reported to present with a largely self-limiting vasculopathy akin to the Kawasaki disease [53]. The pathological and clinical manifestations of this disease include conjunctivitis, keratitis, intermediate uveitis, and inflammation of the optic nerve head [54].

Reyes-Bueno et al. detailed the post-COVID-19 Miller Fisher syndrome (MFS) reported in some patients which included complaints of blurred vision and extra-ocular muscle palsy [55]. Neophytou et al. published a systematic review of this syndrome in COVID-19 patients with a conclusion that the condition occurred more frequently after infection as compared to after vaccination [56]. The prognosis of MFS is spontaneous and very good even without administering intravenous immunoglobulin [56].

Generally, viral diseases are directly linked to reduction in CD4+ and CD8+ counts. This in turn has been implicated in the incidence and reoccurrence of diseases such as her-pes-zoster ophthalmicus, infection induced uveitis and infective cellulitis [57].

## 4. Precautions Taken in Ophthalmic Practices to Prevent the Spread of the Virus

Standard global precautions were advocated worldwide to stem the spread of the disease at the onset of the pandemic, including handwashing [58], using face masks [59], social distancing [60], coughing into the elbow [61], etc. Face masks with breath shields were found to reduce cough droplet spread by 99.98% in an eye care setting [62]. This was in comparison to breath shields alone, which reduced cough droplet propagation by 99.93% as compared to coughing without any protective barrier. ALBalawi reviewed clinical guidance from various regulatory bodies on the safe re-opening of eye hospitals after the lockdown [63]. He advocated a system similar to what Repici et al. [64] recommended, namely classifying patients as shown in Table 2.

Generally, walk-in visits to clinics and hospitals significantly dropped at the beginning of the pandemic and are just starting to reach pre-pandemic levels [65]. One strategy was to rearrange the daily clinics and sub-specialties for better efficiency. Surgical and post-surgical consultations and visits were prioritized while non-surgical visits were postponed [66].

Cornea surgery policies were also updated to reflect that surgery be stopped for patients presenting with an intermediate to high risk of COVID-19 exposure [67]. This was even as COVID-19 had been isolated from the corneas of infected patients [68].

Contact lenses are common optical devices used mainly in the correction of ametropia amongst others. Reports published have suggested that users transition from contact lenses to spectacle lenses to prevent hand-to-eye contact [69,70]. Furthermore, glasses can be a barrier to airborne viruses [71].

Telemedicine and artificial intelligence (AI) became increasingly important in ophthalmology, as well as other medical specialties. Technology advancements have enabled the development and use of new methodologies and approaches, allowing ophthalmologists to offer eye care and give ophthalmology education and training to patients remotely in specific scenarios [72,73]. Telemedicine and teleophthalmology in particular involve massive investments in diagnostic equipment and personnel who interpret these results [74].

## 5. Adverse Eye Reactions after Vaccination

COVID-19 vaccine development and distribution has been critical in battling the continuing epidemic. As with any vaccination, it is critical to properly assess their safety profile, including any potential adverse effects. The sudden appearance of COVID-19 in late 2019, combined with its lethality, resulted in a worldwide scramble for possible remedies and cures. In the USA, Operation Warp Speed was commissioned to fast track the development of a vaccine [75,76].

There have been a few reports of minor and infrequent ocular (eye related) side effects linked to the COVID-19 vaccine, which tend to be infrequent and can be similar to COVID-19 ocular infections. These symptoms include eye discomfort and pain, pink eye or conjunctivitis, tearing or eyes that water, and minor visual disturbances. The ocular signs have been reported to appear up to 42 days after the vaccination, and the possible cause is vaccine-induced immunologic reactions [77]. When present, these symptoms are usually mild, not serious, and tend to resolve spontaneously.

Rare yet important side effects of COVID-19 vaccines have been reported to include central serous retinopathy, acute macular neuroretinopathy, thrombosis, new-onset Graves’ disease, multiple evanescent white dot syndrome, uveitis, and Vogt–Koyanagi–Harada disease reactivation [78]. The majority of studies in the literature now use retrospective case series or lone case reports, which by their very nature are incapable of proving connection or causality. The etiology of the detrimental effects on the eyes following COVID-19 vaccination may entail the immune system’s reaction to the vaccine.

While the majority of COVID-19 vaccine side effects have been moderate and temporary, there have been a few cases of eye adverse reactions following immunization. It is critical to emphasize that the overall advantages of COVID-19 immunization outweigh the risks of ocular side effects, which are uncommon and usually self limiting. Some of the side effects and associations reported in the literature include the following:

Blurry vision: Blurred vision has been reported as a symptom after COVID-19 vaccination [79]. Nasiri et al. reviewed a cohort of publications totaling 1021 patients, of whom 8.2% reported blurry vision [80]. However, it is noted that blurry vision is a regular complaint in individuals pre- and post-COVID-19.

Conjunctivitis: Several studies have shown incidences of conjunctivitis after receiving the COVID-19 vaccine. Nyakerh et al. found that 33.3% of vaccinated people in their sample had moderate conjunctivitis within 7 days of immunization [81]. Notably, the vaccinated cohort had the same rate of conjunctivitis as the general population. Redness, itching, tearing, and a foreign body sensation in the eye are all signs of vaccine-associated conjunctivitis. Without therapy, these symptoms usually resolve in a few days.

Uveitis refers to inflammation of the uvea, which comprises the iris, ciliary body, and choroid. While uveitis is uncommon, a few instances have been observed following COVID-19 immunization. Bolleta et al., for example, documented a cohort of 42 eyes with uveitis occurring at different times after different COVID-19 vaccinations, including BNT162b2 mRNA, ChAdOx1 nCoV-19, mRNA-1273, and Ad26.COV2 [82]. Similarly, Patel et al. reported on occurrences of posterior uveitis after receiving both virus-based and mRNA-based vaccines [83]. Uveitis caused by COVID-19 vaccinations is usually self-limiting and treatable with topical corticosteroids and cycloplegic medications [84].

Other Ocular Side Effects: Other ocular side effects have been documented following COVID-19 immunization, in addition to conjunctivitis and uveitis, albeit they are relatively rare. Episcleritis [85], scleritis [86], optic neuritis [87], burning sensation [88], itchy eyes [89], and retinal artery blockage are examples of these [90]. Akbari and Dourandeesh described the occurrence of mucormycosis in a COVID-19 patient [91]. These eye adverse effects have been documented in sporadic cases [92], and no definitive link to COVID-19 vaccination has been established.

## 6. The Impact of the Pandemic on Patients, Clinicians, and the Eye Care System as a Whole

The breakout of the new COVID-19 pandemic in early 2020 had a significant social and economic impact on the planet [93]. Many governments instituted extraordinary population restrictions, and many nations required a comprehensive redesign of their various national health systems [94]. To limit the spread of the virus and relieve the burden on national health systems, social distancing and other preventative measures, as well as national lockdown periods, were implemented in a variety of methods in various nations at various times.

Several international health organizations, notably the WHO, advocated broad preventive measures to minimize viral spread [95]. The major ophthalmological societies worldwide immediately developed guidelines to prevent COVID-19 transmission in both students, patients, and healthcare workers in ophthalmic practice [96,97] as eyecare providers have been shown to be at high risk of contracting the SARS-CoV-2 infection.

Being in close proximity to the patient, particularly during slit lamp examination and direct ophthalmoscopy, which increases exposure to respiratory droplets; potential contact with infected nasal, oral, and ocular secretions; daily interaction with a large number of patients; and frequent engagement with a specific population subgroup, such as older individuals who are at a higher risk of having underlying health co-morbidities, are all reasons for being classified as high risk [98,99]. The following scenarios were reported in ophthalmology practices in the literature:

Decrease in patient volume: During the pandemic, several ophthalmology clinics saw a significant decrease in patient visits. This can be ascribed to variables such as lockdown measures, travel limitations, fear of viral exposure, and the allocation of healthcare resources to COVID-19 cases [100].

Limited access to care: Some patients faced challenges in accessing ophthalmology clinics due to restrictions on movement especially in countries with large numbers of fatalities [101]. This was particularly prevalent during the initial stages of the pandemic when healthcare facilities were focused on managing COVID-19 cases.

Delayed diagnosis and treatment: A decrease in patient numbers and visits may have resulted in delays in the diagnosis and treatment of eye problems. Timely intervention is critical for certain disorders, such as retinal detachment, age-related macular degeneration, or glaucoma, to avoid disease progression and maintain vision. These delays may have an effect on patient outcomes and necessitate additional care efforts in the post-pandemic phase [102].

Increased backlog and future demand: As the epidemic receded and healthcare services resumed normal operations, ophthalmology clinics confronted a backlog of appointments and treatments that had been postponed. To clear this backlog and meet the pent-up demand for eye care services, careful planning, resource allocation, and effective scheduling techniques will be required [103].

## 7. The Future of Ophthalmology Conditioned by This Global Pandemic Experience

In the midst of a pandemic, ethical questions about eye care have become critical. These cover crucial topics such as: what are the visual repercussions of postponing visits and elective surgeries? When presented with a COVID-positive patient who requires immediate surgery, what criteria should be considered to weigh the risks of postponing the treatment versus aggravating the patient’s condition or potentially infecting the healthcare team? What are the benefits of reallocating healthcare staff, particularly ophthalmologists, to emergency departments or critical care units (ICUs) [104]?

It can however be agreed that the future of ophthalmology holds significant changes and adaptations following the COVID-19 pandemic. Some potential developments include:

An increase in the application of telemedicine: The pandemic has hastened the implementation of telemedicine in ophthalmology, enabling distant consultations, monitoring, and follow ups. This trend is expected to continue, providing patients with more convenience, fewer travel difficulties, and enhanced access to eye care services [105]. Ophthalmology education will also adjust to the new realities of off-site learning. This will potentially play a big part in knowledge transfer from specialists who ordinarily would not have had the opportunity to interact with doctors and residents across countries and even continents [92,106]. Padmaja et. al. showed that patients were comparatively satisfied with being examined by teleophthalmology as compared to seeing an in-person ophthalmologist [107]. The Muranga Teleophthalmology Study also tested this hypothesis and reported a fair correlation in satisfaction in patients when in-person consultation was compared to remote consultation. Their study however combined age-related macular degeneration patients and diabetic retinopathy sufferers [108]. The pandemic was however not all negative, a key positive was that remote patient care enabled multiple clinicians across long distances to participate in the management of a single patient [109].

Artificial intelligence (AI) integration: AI technologies such as computer vision algorithms and machine learning models have already demonstrated promise in aiding with the diagnosis, screening, and treatment of eye disorders [110]. AI is predicted to play a larger role in ophthalmology in the future, assisting in early identification, individualized treatment strategies, and improved patient outcomes [111]. Remote-controlled robotic surgery may begin to see a greater acceptance among surgeons [112]. Already, some techniques, like funduscopy, have begun to shift towards AI assisted fundus photography [113]. Alafaleq conducted a systematic review on cyber surgery by searching relevant clinical related database. In particular he reviewed major medical databases and concluded that this technology was a remedy to shortage of skilled surgical personnel and long distances [114]. The drawbacks of cyber surgery included the steep costs in setting it up, ethical considerations, and the presence (or absence) of legislations relating to its use in different localities.

Remote diagnostic advancements: In its basic form, a remote diagnostic tool can be as simple as a picture uploaded via the internet from one location to another. Usually, it is sent to a more experienced clinician or system for interpretation. Remote monitoring devices and home-based testing kits are projected to grow more advanced, allowing patients to self-assess their eye health. This can help in the early diagnosis of problems and fast action, which is especially important for people who may have difficulty accessing in-person treatment [115]. Horton et al. set out guidelines for diabetic retinopathy screening and this has been useful in standardizing telediagnostics [116]. Verma et. al. reported on the use of telediagnostics in the imaging of diseases of the orbit and the adnexia. They were able to correctly diagnose adnexia diseases in 101 of a total of 3497 patients and go on to recommended management including surgery [117]. European researchers assessed using a remote algorithm for screening for diabetic retinopathy in the TOSCA study. They reported that using only two images from the centers, they were able to properly grade diabetic retinopathy remotely with a very high correlation [118]. The average turn-around time to complete an off-site assessment was a mere five minutes. Schneider et al. went further to assess the quality assurance principles used during the TOSCA. They found the processes involved in acquiring the images at source and downloading at the interpretation centers to be reliable and reliable [119]. Non-surgical eye care provided by optometrists will also significantly benefit from adoption of remote eye care and diagnostics [120].

Enhanced infection control measures: Infection control, especially among health care workers was crucial in stemming the tide of COVID-19. The early days were especially difficult as many workers had no idea how to protect themselves or lacked the necessary materials to achieve proper self-protection [121]. Frontline health care professionals continue to render care to patients despite documented shortages in personal protective equipment. These shortages disproportionally affected minorities, with women also being affected more than men [122]. Nurses were also likely to experience and report symptoms of depression in situations where they did not have access to personal protective equipment [123]. A study found that 2% of clinical workers caring for COVID-19 patients in a facility ultimately contracted the disease. Furthermore, about 80% of these workers had no proper training on how to handle themselves around COVID-19 patients. This was alarming considering the potential morbidity this disease was capable of [124]. Being above 45 years of age, the presence of comorbid conditions, and being doctors were all risk factors of COVID-19 infection in health care workers [125]. Qureshi et al. conducted a scoping review into the preparedness of health care workers to fight the COVID-19 disease. Their transnational review looked at 14 countries that had reported infection protection and guidance trainings before the pandemic. Surprisingly, the study observed that only four of fourteen countries considered training healthcare workers on infection prevention and control [126]. To reduce the risk of transmission, ophthalmic practices will continue to focus tight infection control methods [127]. Frequent disinfection of contact surfaces is advised for clinical settings including the walls [128]. Eye care workers are also advised to prioritize disposable equipment over reusable ones, i.e., using the TonoSafe disposable probes instead of the reusable probes for tonometry. When it is not possible to obtain disposable equipment and supplies, such equipment should be cleaned after every patient and regularly with 75% ethanol [129]. Having patients sterilize their hands upon entry and also frequently during their stay also prevents transmission of pathogens from hand to surface and vice versa [130]. Modifications to waiting spaces and the use of protective barriers and technology to safeguard the safety of both patients and healthcare workers are other examples of this.

Teamwork and multidisciplinary care for patients and healthcare workers: The pandemic brought to light the necessity of inter-professional teamwork in healthcare [131]. To create comprehensive techniques for addressing ocular symptoms of systemic disorders and enhancing patient care, ophthalmologists may work more closely with other specialists such as infectious disease experts, pulmonologists, and epidemiologists [132]. These adjustments go further than clinical experts and extend to administrative staff who were also in the line of the virus. Durmaz-Engin et. al. showed that ophthalmologists reported significantly increased levels of depression, anxiety, and stress as measured using the DASS-21 psychometric scale during the pandemic [133]. Sunil et. al. reported that there was no statistical difference in depression and stress scores between clinical and nonclinical healthcare workers in their study [134]. Creating plans to prevent burnout (as defined by an exhibition of exhaustion and disengagement from work) will help frontline staff cope with similar outbreaks in the future [135]. The delivery of clinical research is also a key part of many ophthalmology settings which were therefore also affected by the pandemic. Ophthalmologists are finding ways to cope with the demands of clinical research output while maintaining eye care services across the spectrum [136].

Better patient education and awareness will be prioritized: Ophthalmologists will likely place a greater focus on patient education on eye health, preventative techniques, and the significance of frequent screenings. Raising knowledge of the relationship between systemic health and eye diseases can encourage people to take proactive steps to protect their ocular health [137]. This will largely be achieved by augmenting the training of students and staff on the risks and prevention strategies in pandemic situations [138]. It was observed that healthcare systems in developing countries were ill equipped to handle the number of patients occasioned by the pandemic [139]. Governments scrambled to erect hospitals and channel funding to COVID-19 response efforts [140]. Advocacy and political will should be encouraged so that citizen education and infrastructural development is proactive instead of reactive.

Advocacy on safer ocular habits: The COVID-19 virus is known to be easily transmitted via human generated aerosols and survive for up to 4 days on surfaces [141]. Hand to face transmission has been described as one of the common routes of self-inoculation of the COVID-19 virus with humans estimated to touch their face an average of 24–50 times every hour [142,143]. The eyes, nose, and mouth form a “T” that represents the most likely parts for the face to be touched [144]. In a study by Phan et al. [145], healthcare workers attending to COVID-19 positive patients were observed to touch their own faces with ungloved hands 50% of the time. Kantor has suggested that wearing facemasks may potentially increase the frequency of hand-to-face contact as wearers try to relieve the skin irritation occasioned by them [146]. Another study suggested that wearing a face mask did not significantly alter the probability of face touching [147]. It stands to reason that vigorous behavioral adjustment and community awareness advocacy on face touching will contribute to the reduction of this infection transmission route [142].

## 8. Long COVID in Ophthalmology

Standard duration of COVID-19 symptoms have been reported to be between 4–12 weeks [148]. Long COVID refers to the diverse symptomatology that continue long after the average duration acute phases of the disease [149]. Long COVID usually presents with initial symptoms of the disease which may become permanent, occur intermittently, or improve over time [150]. This is usually followed long-COVID sequelae which are lasting tissue injury observed after a minimum of 12 weeks post initial presentation [150]. These could be due to COVID-19 related vasculopathies, immune-related deficits, post-intensive care disease, and multi-systemic inflammatory syndrome [151].

Nasserie et al. [152] conducted a systematic review on the distribution of symptoms related to long COVID. They pointed out the heterogeneity of symptoms across the data sampled as presenting with far reaching implications for health care delivery and general quality of life of the sufferers. Osikomaiya et. al. [153] also reported on an African-based survey of long COVID and arrived at similar conclusions. Jadali and Jalil indicated that ophthalmologic sequelae of COVID-19 may occur many weeks after presentation and remission of other symptoms just as they may appear right at presentation [154]. These late-onset ocular deficits may include any combinations of corneal desensitization and increased dendritic cell density among others. Confocal microscopy has been used to demonstrate that these two conditions are directly related with long COVID [155]. The presence of neurological system was noted as a risk factor [156].

## 9. Conclusions

Like other healthcare professions, COVID-19 presented a probably greater number of challenges and changes than any other singular event in recent history to the ophthalmology community. It is noted that the first doctor to raise an alarm about COVID-19 was an ophthalmologist, who later succumbed to the disease himself at the young age of 34 years [157]. This pandemic had significant economic effects on the eye care industry which is just getting back to its pre-pandemic levels. The lessons learned are, however, important to preparing for the next pandemic. Even though the COVID-19 pandemic is practically over, the world cannot afford to go back to the lows of pre-pandemic healthcare. Handwashing, disinfection of high contact surfaces, and proper sneezing/coughing etiquette are not limited to controlling the spread of the SARS-CoV-2 virus, but indeed for any infective conditions. With the current knowledge of the risk factors for COVID-19, it is prudent for clinical teams to take steps to protect the healthcare workers at the highest risks of infection. Ophthalmology departments must ensure continuous training of clinical and non-clinical staff on universal health care practices to reduce the spread of infection. Governments and private hospital administrators would do well to set up emergency funding and supply lines to ensure needed equipment and supplies are handy in the event of another outbreak. The world need not suffer the same level of disruption to health services as occasioned by a similar outbreak in the future.

## Figures and Tables

**Table 1 microorganisms-11-02220-t001:** Notable variants of the SARS-CoV-2 their origins.

Variant	Origins	Remarks
Alpha variant (B.1.1.7)	First identified in the UK	It has been found to be more contagious than the original strain [12].
Beta variant (B.1.351):	First identified in South Africa	It has mutations that may make it more resistant to some antibodies [12].
Gamma variant (P.1)	First identified in Brazil	It is thought to be more transmissible and may be able to re-infect people who have already had COVID-19 [13].
Delta variant (B.1.617.2)	First identified in India	It is highly transmissible and has become the dominant strain in many parts of the world [14].
Omicron variant (B.1.1.529)	Emerged in South Africa in November 2021	A heavily mutated, highly virulent variant that quickly spread around the world [15].
Epsilon, Zeta, Eta, Theta, Iota, and Kappa variants		Are being closely monitored to understand their differentiating characteristics [16,17].

**Table 2 microorganisms-11-02220-t002:** Classification of patients presenting to the clinic during the COVID-19 pandemic.

Low Risk	This Group Included Individuals Without COVID-19 Symptomatology and Had Not Been in Contact with High-Risk Areas of Positive Patients
Intermediate risk	This group included individuals with COVID-19 symptomatology but had not been in contact with high-risk areas of positive patients OR anyone who had been in contact with a COVID-19 patient or stayed in a high-risk area but who did not have any symptoms.
High risk	This included people who had at least one COVID-19 symptom and had been in contact with a confirmed case or stayed in a high-risk area.

## Data Availability

Not applicable.

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
