# Peer review of "Considerations of COVID-19 in Ophthalmology"

_microorganisms, 2023, doi:10.3390/microorganisms11092220_

Round 1
Reviewer 1 Report
The purpose of this article is to provide an overview of all aspects of COVID-19 in ophthalmology, including ocular manifestations associated with the disease, how SARS-CoV-2 infection is transmitted, precautions to prevent transmission of the virus in ophthalmic practice, drugs and vaccines to treat COVID-19, and the impact of the outbreak on patients, clinicians, and the eye care system as a whole. However, the overall thinking of this paper is not deep enough, the discussion is not thorough, there is a lot of room for improvement.
1.The third part, Ocular manifestations of Covid ‐ 19, does not go far enough. It is suggested to expand this section, which can be divided into two parts: ocular manifestations and confirmed ocular diseases.
2.How can we draw a conclusion“It can however be inferred that if the virus can be isolated from tears and intraocular fluid, then it can also be isolated from other ocular tissue”
3.The title of the fifth part is“Drugs and vaccines used in the treatment of COVID‐19”However, most of them describe adverse eye reactions after vaccination, and it is recommended to change the title;
-
Author Response
The purpose of this article is to provide an overview of all aspects of COVID-19 in ophthalmology, including ocular manifestations associated with the disease, how SARS-CoV-2 infection is transmitted, precautions to prevent transmission of the virus in ophthalmic practice, drugs and vaccines to treat COVID-19, and the impact of the outbreak on patients, clinicians, and the eye care system as a whole. However, the overall thinking of this paper is not deep enough, the discussion is not thorough, there is a lot of room for improvement.
In accordance with the Reviewer, additional pertinent information has been added throughout the text to render each section more complete. The total word count of this last version is now 11,488 words.
1.The third part, Ocular manifestations of Covid ‐ 19, does not go far enough. It is suggested to expand this section, which can be divided into two parts: ocular manifestations and confirmed ocular diseases.
As suggested by the Reviewer, this section has been divided into 2 parts and expanded with several paragraphs to read:
“Ocular manifestations of Covid – 19 and
Confirmed ocular diseases in Covid-19”
- Dry eye diseases are the most commonly reported ocular complications of Covid-19 [51]. Krolo et al. reported that face mask use was related to higher Ocular Surface Disease Index (OSDI) scores [44]. Other possible reasons may include increased screen time occasioned by working from home and the inability to obtain needed dry eye medications due to the lockdown [52].
- Children with Covid-19 have been reported to present with a largely self-limiting vasculopathy akin to Kawasaki disease [53]. The pathological and clinical manifestations of this disease include comprise conjunctivitis, keratitis, intermediate uveitis, and inflammation of the optic nerve head [54].
- Reyes-Bueno et al. detailed the post-Covid-19 Miller-Fisher syndrome (MFS) reported in some patients which included complaints of blurred vision and extra-ocular muscle palsy [55]. Neophytou et al. published a systematic review of this syndrome in Covid-19 patients with a conclusion that the condition occurred more frequently after infection as compared to after vaccination [56]. The prognosis of MFS is spontaneous and very good even without administering intravenous immunoglobulin [56].
- Generally, viral diseases are directly linked to a reduction in CD4+ and CD8+ counts. This, in turn, has been implicated in the incidence and reoccurrence of diseases such as herpes-zoster ophthalmicus, infection-induced uveitis, and infective cellulitis [57].
2.How can we draw a conclusion “It can however be inferred that if the virus can be isolated from tears and intraocular fluid, then it can also be isolated from other ocular tissue”
The Reviewer makes a good point. This sentence has been removed to avoid this unjustified hypothesis.
3.The title of the fifth part is “Drugs and vaccines used in the treatment of COVID‐19” However, most of them describe adverse eye reactions after vaccination, and it is recommended to change the title.
As kindly suggested, the title has been changed to read “Adverse eye reactions after vaccination”. This section has been expanded to provide additional insights into this issue as follows:
“There have been a few reports of minor and infrequent ocular (eye-related) side effects linked to the COVID-19 vaccine, which tend to be infrequent and can be similar to COVID-19 ocular infections. These symptoms include eye discomfort and pain, pink eye or conjunctivitis, tearing or eyes that water, and minor visual disturbances. The ocular signs have been reported to appear up to 42 days after the vaccination, and the possible cause is vaccine-induced immunologic reactions [77]. When present, these symptoms are usually mild, not serious, and tend to resolve spontaneously.
Rare yet important side effects of COVID-19 vaccines have been reported to include central serous retinopathy, acute macular neuro-retinopathy, thrombosis, new-onset Graves' disease, multiple evanescent white dot syndrome, uveitis, and Vogt-Koyanagi-Harada disease reactivation [78]. The majority of studies in the literature now use retrospective case series or lone case reports, which by their very nature are incapable of proving connection or causality. The etiology of the detrimental effects on the eyes following COVID-19 vaccination may entail the immune system's reaction to the vaccine.”
We thank the Reviewer for a very thorough evaluation of our manuscript. Excellent points have been raised that can improve our manuscript. Modifications have been made based on the issues raised. We have tried to address each point, which can be viewed with the track changes throughout the manuscript. Additional specific issues can be modified upon request.
Reviewer 2 Report
This is an exciting review of the impact of COVID-19 on ophthalmology. The authors should answer the following questions before publication.
What is the impact of vaccination on the eye? Are there written adverse effects of vaccines?
The condition of persistent COVID has been described. Are there persistent ophthalmologic symptoms of COVID?
One aspect that should be discussed is whether involvement through the eyes is possible due to previous infection with surfaces on which the virus is present. ( It should be taken into account that it is described in the literature that people touch their faces between 20 and 50 times every hour.)
Another question related to the previous one is the techniques that should be used to disinfect surfaces and equipment in ophthalmology practices.
Author Response
This is an exciting review of the impact of COVID-19 on ophthalmology. The authors should answer the following questions before publication.
We are grateful to the Reviewer for the positive comments about our review.
What is the impact of vaccination on the eye? Are there written adverse effects of vaccines?
To address this issue, the following has been added:
“There have been a few reports of minor and infrequent ocular (eye-related) side effects linked to the COVID-19 vaccine, which tend to be infrequent and can be similar to COVID-19 ocular infections. These symptoms include eye discomfort and pain, pink eye or conjunctivitis, tearing or eyes that water, and minor visual disturbances. The ocular signs have been reported to appear up to 42 days after the vaccination, and the possible cause is vaccine-induced immunologic reactions [77]. When present, these symptoms are usually mild, not serious, and tend to resolve spontaneously.
Rare yet important side effects of COVID-19 vaccines have been reported to include central serous retinopathy, acute macular neuro-retinopathy, thrombosis, new-onset Graves' disease, multiple evanescent white dot syndrome, uveitis, and Vogt-Koyanagi-Harada disease reactivation [78]. The majority of studies in the literature now use retrospective case series or lone case reports, which by their very nature are incapable of proving connection or causality. The etiology of the detrimental effects on the eyes following COVID-19 vaccination may entail the immune system's reaction to the vaccine.”
The condition of persistent COVID has been described. Are there persistent ophthalmologic symptoms of COVID? persistent ophthalmologic symptoms of COVID
The Reviewer raises an interesting point here, especially considering the clinical importance of long COVID. To address this issue, the following has been added:
“Standard duration of Covid-19 symptoms have been reported to be between 4-12 weeks [148]. Long-Covid refers to the diverse symptomatology that continues long after the average duration acute phases of the disease [149]. Long-Covid usually presents with initial symptoms of the disease which may become permanent, occur intermittently, or improve over time [150]. This is usually followed Long-Covid sequelae which are lasting tissue injuries observed after a minimum of 12 weeks post-initial presentation [150]. These could be due to Covid-19 related vasculopathy, immune-related deficits, post-intensive care disease, and multi-systemic inflammatory syndrome [151].
Nasserie et al. [152] conducted a systematic review of the distribution of symptoms related to Long-Covid. They pointed out to the heterogeneity of symptoms across the data sampled as presenting far-reaching implications for health care delivery and the general quality of life of the sufferers. Osikomaiya et. al. [153] also reported on an African-based survey of Long-Covid and arrived at similar conclusions. Jadali and Jalil indicated that Ophthalmologic sequelae of Covid-19 may occur many weeks after the presentation and re-mission of other symptoms just as they may appear right at presentation [154]. These late-onset ocular deficits may include any combinations of corneal desensitization and increased dendritic cell density among others. Confocal microscopy has been used to demonstrate that these two conditions are directly related to Long-Covid [155]. The presence of neurological system was noted as a risk factor [156].”
One aspect that should be discussed is whether involvement through the eyes is possible due to previous infection with surfaces on which the virus is present. ( It should be taken into account that it is described in the literature that people touch their faces between 20 and 50 times every hour.)
In accordance with the clinically important suggestions made by the author, the following has been added to read:
“ • Advocacy on safer ocular habits: The Covid-19 virus is known to be easily transmitted via human-generated aerosols and survive for up to 4 days on surfaces [141]. Hand-to-face transmission has been described as one of the common routes of self-inoculation of the Covid-19 virus with humans estimated to touch their face an average of 24 - 50 times every hour [142-143]. The eyes, nose, and mouth form a “T” that represents the most likely parts for the face to be touched [144]. In a study by Phan et al. [145], healthcare workers attending to Covid-19 positive patients were observed to touch their own faces with ungloved hands 50% of the time. Kantor has suggested that wearing facemasks may potentially increase the frequency of hand-to-face contact as wearers try to relieve the skin irritation occasioned by them [146]. Another study suggested that wearing a face mask did not significantly alter the probability of face touching [147]. It stands to reason that vigorous behavioral adjustment and community awareness advocacy on face touching will contribute to the reduction of this infection transmission route [142].”
Another question related to the previous one is the techniques that should be used to disinfect surfaces and equipment in ophthalmology practices.
To address the issue regarding disinfection techniques, the following sentences have been added:
“To reduce the risk of transmission, ophthalmic practices will continue to focus on tight infection control methods [127]. Frequent disinfection of contact surfaces is advised for clinical settings including the walls [128]. Eye care workers are also advised to prioritize disposable equipment over reusable ones i.e. using the TonoSafe disposable probes instead of the reusable probes for tonometry. When it is not possible to obtain disposable equipment and supplies, such equipment should be cleaned after every patient and regularly with 75% ethanol [129]. Having patients sterilize their hands upon entry and also frequently during their stay also prevents transmission of pathogens from hand to surface and vice-versa [130].”
We are grateful to the Reviewer for the helpful points provided in the review of our manuscript. We hope that all issues have been addressed in an appropriate manner.